# Bimodular Antiparallel G-Quadruplex Nanoconstruct with Antiproliferative Activity

**DOI:** 10.3390/molecules24193625

**Published:** 2019-10-08

**Authors:** Olga Antipova, Nadezhda Samoylenkova, Ekaterina Savchenko, Elena Zavyalova, Alexander Revishchin, Galina Pavlova, Alexey Kopylov

**Affiliations:** 1Chemistry Department, Lomonosov Moscow State University, Leninskiye Gory 1–40, 119991 Moscow, Russia; zlenka2006@gmail.com (E.Z.); kopylov.alex@gmail.com (A.K.); 2Institute of Gene Biology, Russian Academy of Sciences, Vavilova 34/5, 119334 Moscow, Russia; samoylenkova.n@gmail.com (N.S.); savhenko61@mail.ru (E.S.); revishchin@mail.ru (A.R.); lkorochkin@mail.ru (G.P.); 3Federal State Autonomous Institution (N.N. Burdenko National Scientific and Practical Center for Neurosurgery) of the Ministry of Health of the Russian Federation, 1st Tverskaya-Yamskaya 13/5, 125047 Moscow, Russia;; 4Sechenov First Moscow State Medical University, Institute of Molecular Medicine, B. Pyrogovskaya 2/6, Moscow 119992, Russia

**Keywords:** DNA G-quadruplex oligonucleotide, nanoconstruct, antiproliferative activity, human cancer cell lines

## Abstract

Oligonucleotides with an antiproliferative activity for human cancer cells have attracted attention over the past decades; many of them have a G-quadruplex structure (GQ), and a cryptic target. In particular, DNA oligonucleotide HD1, a minimal GQ, could inhibit proliferation of some cancer cell lines. The HD1 is a 15-nucleotide DNA oligonucleotide that folds into a minimal chair-like monomolecular antiparallel GQ structure. In this study, for eight human cancer cell lines, we have analyzed the antiproliferative activities of minimal bimodular DNA oligonucleotide, biHD1, which has two HD1 modules covalently linked via single T-nucleotide residue. Oligonucleotide biHD1 exhibits a dose-dependent antiproliferative activity for lung cancer cell line RL-67 and cell line of central nervous system cancer U87 by MTT-test and Ki-67 immunoassay. The study of derivatives of biHD1 for the RL-67 and U87 cell lines revealed a structure-activity correlation of GQ folding and antiproliferative activity. Therefore, a covalent joining of two putative GQ modules within biHD1 molecule provides the antiproliferative activity of initial HD1, opening a possibility to design further GQ multimodular nanoconstructs with antiproliferative activity—either as themselves or as carriers.

## 1. Introduction

G-quadruplex (GQ) folded oligonucleotides are capable of inhibiting proliferation of cancer cells, both as themselves and as carriers (see reviews [1,2,3]). Despite knowing these effects for more than a decade, a tentative molecular mechanism is not known yet—even for the most studied case, DNA GQ AS1411 [1,4,5]—most likely because of a cryptic target. The functional effect of oligonucleotide could be a result of many different events: Binding and modulating cell membrane functioning, penetrating a cell, or specific binding and blocking of functionally important GQ-binding proteins inside a cell, thereby yielding a pleiotropic effect. Some events had been experimentally described, such as blocking the signal transduction, for example [6,7,8,9,10].

15-nucleotide DNA aptamer HD1 was originally selected as thrombin binding aptamer [11], and in the current literature it is normally abbreviated as TBA, despite its original name being HD1. Later, an antiproliferative activity of this 15-meric DNA had been discovered [12]. In this publication we will be using the original name HD1, because only antiproliferative properties of this GQ are under study —keeping aside its ability to bind thrombin. The HD1 folds into a minimal chair-like monomolecular antiparallel GQ structure. The HD1 structure is two stacked G-quartets with a potassium cation coordinated in the central cavity [13,14]. Coming back to the thrombin story, both destabilization and excessive stabilization of GQ framework significantly reduces the inhibitory activity of the aptamer, most likely via conformational changes in the two TT-loops that interact with thrombin [14].

In the study of Bates, antiproliferative activity of HD1 was discovered [12]. This activity is compared, for example, with one of the AS1411, which is the current leader of antiproliferative GQs [1,5]. Since then, some occasional conditions and a limited number of cell lines were tested. However, no standard conditions have been established yet, therefore not allowing for any uniform results nor any comparative studies to be made.

Treatment of HeLa cells with 10 μM HD1 for seven days has decreased the cell viability to 35% to 60% [15,16]. Antiproliferative activity of 10 μM HD1 has been shown for Calu-6 lung cancer cells: Treatment for one day does not decrease the cell viability; for two days, the cell viability decreases to 60%; and for three days, it decreases down to 30% [17]. For colon cancer cells HCT116^-/-p53^ harboring a mutation of the p53 gene, treatment with 10 and 50 μM HD1—even for one day—has a notable effect, as the cell viability decreases to 80% and 25%, respectively [18]. For the majority of studies on the antiproliferative effect of HD1, a 10 μM concentration was used. We therefore used this concentration as the ’standard’.

There are a few examples when molecular constructs assembled from separate functional modules retain activity corresponding to the original modules [19,20]. The strategy of assembling multimodular constructs by covalent binding of separate functional modules is used, for example, to create conjugates of aptamers with miRNA that have approached a stage of clinical trials [21,22].

Therefore, because HD1 has definite antiproliferative activity, then it is of value to use this GQ module to build bimodular constructs. A minimal version of this construct is a ‘tween’, made by covalently linking two GQs via a single thymidine residue, producing 31-mer biHD1. Coming back to the story with thrombin, this ‘tween’ molecule had been thoroughly tested for anti-thrombin functioning under the name RA-36 [23], which has high enough functional activity [24]. A tentative structure of RA-36 has two asymmetrical GQs, where the second module is not an ideal GQ [25,26]. Recently, we learned that the bimodular construct could be made by joining HD1 modules with artificial non-nucleotide linker [27]. Assuming that different parts (pharmacophores) of the same GQ framework of HD1 perform two activities—anti-thrombin and antiproliferative ones—it is impossible to predict the antiproliferative activity of biHD1 in advance.

In addition to biHD1 alone, a series of biHD1 derivatives were tested to find a structure-activity relationship: Whether any GQ alterations would affect antiproliferative activity.

For studies regarding antiproliferative activity of some GQs and HD1, only a few cell lines were applied. Generally, an antiproliferative effect depends on both GQ structure and cell line. Therefore, for the particular study of biHD1, a selection of the proper cell line is required. In addition, a lack of standard conditions for testing has hindered the comparison on published data. Therefore, this study is dedicated to the screening of antiproliferative activity of the bimodular DNA oligonucleotide, biHD1 and derivatives for eight human cancer cell lines—including non-malignant human embryonic fibroblasts as a control—to establish an appropriate model for further studies.

## 2. Results

### 2.1. Oligonucleotide Design in Brief

The nucleotide sequences of HD1, biHD1 and its derivatives are shown in Table 1. A more detailed description of the design of biHD1 and derivatives will be provided in the ‘Discussion’ section.

The oligonucleotide biHD1 is a 31-mer, and it was designed by covalently joining two 15-meric HD1s via single thymidine residue. biHD1 was originally proposed as an anti-thrombin aptamer under the name RA-36 [23]. To study antiproliferative activity, in particular, we have focused on the correlation of conformation of the GQ and activity, seeing as anti-thrombin activity and antiproliferative activity are supported by different parts and/or pharmacophores of the HD1 molecule [16,17,18]. The following series of derivatives were designed; firstly, biHD1-C3 possibly has a reduced reciprocal influence of two GQs; secondly, biHD1-T4A,T20A possibly has less compact GQ structures; thirdly, biHD1-5’-Δ2G possibly does not have the 5′-terminal GQ; and fourthly, biHD1+Ba possibly has an unusual conformation of GQs [25].

### 2.2. Monitoring of biHD1 and Derivatives Conformations

Interactions of two GQs with each other within biHD1 molecule has been indirectly demonstrated previously [28].

Circular dichroism spectroscopy is a straightforward conventional approach to monitor both GQ folding and its orientations because of highly characteristic spectra of GQ. The CD spectra of biHD1 and its derivatives have shown a topology of antiparallel GQ similar to single-modular HD1—except for biHD1+Ba (Figure 1). The data have supported the possibility of covalently joining two GQs with a single T-nucleotide. Conformation of the initial GQ-modules is not essentially distorted.

The molar circular dichroism for biHD1 with K^+^ cation is 1.5 times more than that for HD1, and it has proved the existence of two unequal GQs [25,26].

Replacement of T-nucleotide as a linker with more flexible three-methylene bridge led to a 1.3-time decrease of maximum 294 nm, showing some deviations from the initial conformation of HD1. A decrease of the maximum for two other derivatives correlates with expected distortions of the GQs of biHD1. A decrease of 1.3 times or slightly more of the maximum happened when two base-paired thymines of lateral TT-loops had been replaced with two more bulky adenines in biHD1-T4A,T20A. Several other structural distortions were observed for biHD1-5’-Δ2G with a 1.7 times decrease of the maximum when the first GQ had been destroyed.

biHD1+Ba shows both an increased maximum of the molar circular dichroism and bathochromic shift; its spectrum correlates well with spectra of single-modular HD1 with Ba^2+^ [25]. The molar circular dichroism for biHD1 is two times more than that for HD1, and it has proved the existence of two GQs that are more proper than in case of K^+^ cation. An increase in the amplitude of the maximum of the CD spectrum and better proportion could be a result of conformational freezing of GQs due to stronger chelation of the guanine bases with barium cations then that with potassium one [29].

### 2.3. Antiproliferative Activity 

#### 2.3.1. Antiproliferative Activity of biHD1

To reveal the dose-dependent cellular response, three concentrations of biHD1 were tested: 0.1, 1.0 and 10 μM. Unsurprisingly, the responses for different cell lines turned out to be different, which suggests that the mechanisms of biHD1 action are not uniform but depend on the cell line nature (Table 2, Figure 2A–C, Appendix A).

The most pronounced effect of the bimodular biHD1 on the cell viability was for lung cancer cells RL-67 (Figure 2A). The effect was dose-dependent; the treatment with 0.1 μM biHD1 decreased the cell viability to 70%, while the treatment with 10 μM biHD1 decreased it further from 30% to 40%.

The U87 cell line showed approximately the same dose-dependent response when treated with biHD1 (Figure 2B). All other cell lines were much less sensitive to a low concentration of biHD1, showing measurable effects only when the oligonucleotide was at the ’standard‘ of 10 μM concentration (Appendix A, Table 2). Finally, the non-malignant cell line of human embryonic fibroblasts is weakly sensitive to treatment with biHD1, and a decrease in the cell viability was about 10% with 10 μM biHD1 (Figure 2C).

The antiproliferative activity of biHD1 was independently tested by assessing the change in the number of proliferating cells for U87 in particular, and MCF7 as a control, treated with solutions of oligonucleotide biHD1 of 0.1, 1 and 10 μM for 72 h. Figure 3 shows that after incubation of U87 with 10 μM biHD1 for 72 h, there is a decrease in a number of Ki-67 positive cells (green versus blue, Figure 3, four upper fields), and there are only a minor changes for the MCF7 line (green versus blue, Figure 3, four bottom fields).

#### 2.3.2. Antiproliferative Activity of biHD1 Derivatives

To test the hypothesis that GQ structure is responsible for the antiproliferative activity, a series of biHD1 derivatives were created (Table 2). In Section 2.1., “Oligonucleotide Design in Brief”, we introduced a rationale for modulation a structure of GQs of biHD1 to different extents. The three-methylene bridge within biHD1-C3 makes two GQ of initial biHD1 more independent, possibly reducing reciprocal effects. It decreased the antiproliferative effect of biHD1-C3 (Figure 2D–F and Appendix A). For the sensitive cell line, RL-67, the effect of biHD1-C3 is smaller than for the original biHD1: 60% versus 40% of cell viability, respectively; and the effect could be seen only at high concentrations of biHD1-C3—10 µM. For several cell lines, U87, HCT116, MCF7 and mS, the response was nullified (Figure 2E and Appendix A).

If the more flexible biHD1-C3 molecule has decreased antiproliferative potential, then it is worth to directly compare its activity with properties of single-modular HD1.

Until now, an antiproliferative effect of single-modular GQ HD1 has been shown on the limited number of cell lines such as HeLa, Calu-6 and mutant HCT116^p53-/-^ [15,16,17,18]. In particular, U87 cell lines and embryonic human fibroblasts have not been tested for the antiproliferative activity of HD1. We have directly compared the antiproliferative effect of single-modular GQ HD1 and bimodular GQ biHD1-C3 by MTT test after 72 h of treatment (Figure 2E,F and Appendix A). Viability of U87 cells had not decreased when treating with HD1 and biHD1-C3—contrary to biHD1 with 45% of survival. The human embryonic fibroblast cell line was weakly sensitive to 10 μM HD1 treatment with a 10% decrease in the cell viability.

To introduce possible minor deviations in the GQ structure of biHD1, TT-loops structure/interactions had been changed. In each GQ TT-loops could interact via TT base pair T4-T13 in the 5′-terminal GQ and T20-T29 in the 3′-terminal GQ. In biHD1-T4A,T20A derivative, T4 and T20 are replaced with adenine residues to change the nature of the base pair partner. These replacements eliminate the antiproliferative activity of biHD1-T4A,T20A for all cell lines—except for the most sensitive lung cancer RL-67 cell line with 50% cell viability (Figure 2G–I and Appendix A). For all concentrations of biHD1-T4A,T20A, nearly no effect on human embryonic fibroblast cells was observed.

Oligonucleotide biHD1-5’-Δ2G could only have a single 3’-terminal GQ, because 5′-terminal GQ could not be folded due to the deletion of two 5′-terminal critical G-residues. BiHD1-5’-Δ2G does not exhibit antiproliferative activity for all cell lines—except for U87 (Figure 2J–L and Appendix A).

The complex of biHD1 with barium cation, biHD1+Ba, has an unusual conformation of both GQs [25]. Interestingly enough, biHD1+Ba does not influence the proliferation of any cell line under study (Appendix A). Barium cation itself could decrease cell viability of some cell lines [30]; therefore, in control experiments, a cytotoxic effect of barium cation was evaluated (Appendix A). Only weak cytotoxic effect of barium cations on RL-67 cell line and fibroblasts have been found, and it is eliminated when barium cation is chelated within biHD1+Ba complex.

## 3. Discussion

Antiproliferative activity of DNA oligonucleotide with G-quadruplex structure (GQ) is well known for more than decades. Bates et al. developed an antiproliferative GQ oligonucleotide AS1411 and made a thorough study aiming to translate AS1411 into the clinic [1,4,5]. Then, HD1, another GQ structure with antiproliferative activity, was discovered [12]. Since then, some other GQ molecules have been identified [2,3], and for some of them IC_50_ has been determined [3]. Nevertheless, even for the widely studied AS1411, both the exact target and mechanism of action are not yet known [1]. 

Several strategies for further molecular design could be proceeded either by irrational random selection or rational design. A simple way for rational design is to reveal some minimal functional structures and manipulate them; for example, to assemble them covalently into multi-functional constructs. The creation of multi-modular structures combining several structural or functional modules within a single molecule appears to be a promising approach for drug development.

A simple example of antiproliferative GQ is a chair-like monomolecular antiparallel GQ, 15-mer HD1 [12]. In some cases, even modified versions of HD1 still exhibit antiproliferative activity [16,17,18]. Therefore, aiming to explore a possibility to use HD1 as a module for creating more-complex molecules, we decided to make the simplest construct of biHD1 by covalently linking two GQ modules via a single thymidine residue. This ‘tween’ molecule could lose structure/activity or save structure/activity either completely or with some deviations.

This construct has already been made for other purposes [23], but its structure has not yet been established in detail; it is assumed that the existence of two slightly unequal GQs with a topology of monomolecular antiparallel for each of them are similar to the GQ topology of HD1 [26,28].

Circular dichroism spectroscopy is an ideal method to monitor GQ folding, because every particular topology of GQ has a characteristic spectrum, and the value of molar circular dichroism is proportional to a percentage of GQ fold in the sample. As was shown before in [26], CD spectrum of biHD1 in 5 mM KCl has maxima at 294 and 247 nm and minima at 267 and 230 nm, which completely correlates with CD spectrum of initial HD1, but the amplitude of the maximum at 294 nm is higher. In the current study, a comparison of CD was made at 10 mM KCl, and it turned out that the maximum of biHD1 is 1.5 times higher than for HD1—which perfectly indicates a folding of additional GQ. The more pronounced difference of values is due to the strict dependence of GQ folding on KCl concentration (see, for example [25]). CD melting data have proved the stability of GQ structures of biHD1 under conditions of antiproliferative activity testing (Appendix A). 

After GQ structure of biHD1 had been proved, the next question is whether HD1 retains its antiproliferative activity. Antiproliferative activities of GQs with different topology vary; there is no molecular mechanism of its activity. Therefore, the very first step to check an antiproliferative potential of biHD1 is to screen the sensitivity of different human cancer cell lines and select suitable ones for further research. Different human cancer cell lines: Melanoma, glioblastoma; lung, breast, intestines and prostate cancer were used in this screening (Table 3). Some cell lines, e.g., HeLa and HCT116 mutant, have already been used to check the antiproliferative effect of HD1 [15,16,17,18].

The antiproliferative effect depends on the concentration of GQ, the duration and frequency of treatment. In many cases, the most pronounced effect has been found at high GQ concentration of 50 μM [16,17,18] and a very long duration of 7 days [15] without intermediate treatment. Therefore, for the primary testing in this study, a low concentration range 0.1–10 μM, of biHD1, with a treatment duration of three days, was selected.

It is generally accepted that the cell viability correlates well with proliferating potential, and that the conventional approach is the MTT test [31]. The MTT test measures cell viability due to the metabolic activity, by a reduction of the water-soluble, yellow tetrazole salt (MTT) into the insoluble dark blue formazan. The amount of reduced MTT is proportional to the number of living cells. MTT is a simple and straightforward approach for sensitivity screening of different sets of both oligonucleotides and cell lines. More valuable information on antiproliferative information could be obtained by direct measuring the amount of proliferating cells, and this was performed for some particular samples after selecting sensitive cell line, U87.

Antigen KI-67 (also known as Ki-67 or MKI67) is a nuclear protein that is strictly associated with cell proliferation, but is absent in resting (quiescent, G0) cells; and hence is a classical cellular marker for proliferation [32,33]. Ki-67 is an excellent marker to measure the growth fraction of a given cell population, identified by monoclonal antibody Ki-67. Even more, the fraction of Ki-67-positive tumor cells (the Ki-67 labeling index) is often correlated with the clinical course of cancer. Therefore, by comparing numbers of proliferating cells (stained with fluorescent Ki-67-antibody, green images) with a total number of cells (stained with bisbenzimide nuclear fluorescent dye, blue images) one could directly estimate an antiproliferative effect of biHD1.

### BiHD1

According to the MTT data, biHD1 has antiproliferative activity for all tested human cancer cell lines. It is the most active for two cell lines: Lung cancer cells RL-67 and CNS cancer cells U87, with a cell viability of less than 50%, and it is important that the response is dose-dependent (Figure 2A,B and Appendix A, Table 2). Three cell lines with cell viability from 50% to 80% are HeLa (as commonly used), colon cancer cells HCT116, and breast adenocarcinoma cells MCF7. Three cell lines have a rather high cell viability from 80% to 100%: Prostate cancer cells PC3, melanoma cells mS, and non-cancerous embryonic fibroblasts cell line hEF. It is important to stress that biHD1 does not affect the viability of non-malignant fibroblast cell line (Figure 2C).

Two examples illustrate the idea that there is no uniform mechanism of action of different GQs for different cell lines. Firstly, the cell viability of human embryonic fibroblast cells has decreased to 70% after treatment with 10 µM biHD1-C3 (Figure 3B), and the original biHD1 has no effect. Curiously enough, biHD1-C3 increased proliferation of PC3 prostate cancer cells up to 130% in a dose-dependent way (Appendix A). 

The MTT assay is a colorimetric assay to measure cell metabolic activity, and not proliferation itself. Nevertheless, the assay is widely applicable to estimate cytotoxicity (loss of viable cells) or cytostatic activity (shift from proliferation to quiescence) of potential anti-cancer agents, because all these parameters are interlinked. Therefore, it could be more informative to evaluate the antiproliferative activity of biHD1 by an independent method, for example, by direct counting of proliferating cells before and after treatment with biHD1. 

We focused on the sensitive U87 cell line and compared with the less sensitive MCF7 cell line. Immunohistochemistry results indicated that the number of Ki67-positive cells, indicating proliferating cells, was decreased in the case of U87 cell line (Figure 3) treated with 10 μM biHD1. This is not the case for less sensitive MCF7 cells.

To test a hypothesis, which is that the GQ structure could be responsible for the antiproliferative activity, a series of biHD1 derivatives were created (Table 1).

### BiHD1-C3

The two GQs within the original biHD1 are linked via a single thymidine residue, providing a rather compact structure that unfolds in a rather cooperative manner [28]. On the other hand, the antiproliferative activity could be provided either by the entire structure or by an individual GQ itself. Trying to resolve this ambiguity, we made biHD1 derivative, biHD1-C3, with a more flexible linker between two GQs—the propanediol residue. In this case, rather rigid nucleoside moiety is replaced with a flexible three-methylene bridge. The CD spectrum (Figure 1) confirms the idea that a more flexible linker in biHD1-C3 disturbs the original conformations of GQs as compared to biHD1, in which the structures of the two GQs appear to have a reciprocal effect. Thus, biHD1-C3 could have more autonomous GQs. This could affect biHD1-C3 activity, for example, both for the ability to penetrate the cell and for interactions with GQ-binding proteins inside the cell. Indeed, the antiproliferative activity of biHD1-C3 had changed for all cell lines (Figure 2D–F and Appendix A). In general, biHD1-C3 became much less toxic to the cancer cell lines, although the very sensitive lung cancer cell line RL-67 still shows a decrease in the cell viability—but only at high concentration of biHD1-C3, 10μM. In contrast to the original biHD1, biHD1-C3 slightly reduced the viability of control fibroblast cells to 70% at high concentration; the effect was dose-dependent (Figure 2F).

The most striking example of unusual activity of biHD1-C3 is its effect for the PC3 cell line, where proliferation is even increased by 30% after treatment with 10 μM biHD1-C3. This activity is dose-dependent (Appendix A). Similar examples of increased cell viability after the addition of an oligonucleotide are described for some other GQs [34]. 

The properties of biHD1-C3—having more independent GQs—led us to directly compare the properties of bimodular biHD1-C3 and single-modular HD1.

### HD1

The antiproliferative effect of HD1 is described in the literature just for several types of cell lines, for example, MCF7 [3,12], HeLa [15,16], and mutant HCT116^p53-/-^ [17,18]. We have focused on U87 CNS cancer cells. It turned out that HD1 has no antiproliferative activity, exactly as HD1-C3 (Appendix A)—but not as initial biHD1. Therefore—particularly for the U87 cell line—it is in line with an idea that the entire construct of biHD1 is required for the antiproliferative activity—not just GQ itself. 

### BiHD1-T4A,T20A

The single-module of HD1 has T4-T13 noncanonical base pair, which fixes a mutual conformation of two lateral TT-loops. This T4-T13 pair stacks on the adjacent G-quartet and stabilizes GQ [14]. In biHD1-T4A,T20A derivative, two thymines from both TT-loops, T4 and T20, were replaced with two adenines, producing T-A juxtaposition instead of TT base pair. Due to the obvious different geometry of T-A and T-T, some conformational distortions could be expected. Indeed, the CD spectrum of biHD1-T4A,T20A shows deviations from the original conformation of biHD1 (Figure 1). The possible conformational changes are not obvious, and computer MD modeling is now in progress. In any case, biHD1-T4A,T20A loses antiproliferative activity for almost all cell lines. Again, the only exception is sensitive RL-67 cell line, which exhibits sensitivity, but for the high concentration 10 μM biHD1-T4A,T20A (Figure 2G–I and Appendix A).

### BiHD1-5’-Δ2G

To check the hypothesis that the antiproliferative activity requires a GQ structure of biHD1, biHD1-5’-Δ2G molecule was designed. The 5’-terminal GQ could not fold due to the deletion of critical 5′-terminal two Gs forming GQ. For the single-module of HD1, the lack of the 5′-terminal two Gs could yield an alternative structure under certain conditions (G-triplex, instead of G-quadruplex), but the stability of this conformation is low [35]. Indeed, in the CD spectrum of biHD1-5’-Δ2G, there is a strong decrease in the amplitude of 294 nm (Figure 1). The value is two times lower than for the initial biHD1, but not as low as the value for completely unfolded biHD1 [25]. Not surprisingly, the antiproliferative activity of biHD1-5’-Δ2G is significantly reduced for almost all cell lines (Figure 2J–L and Appendix A). This means that the 5’-terminal GQ module plays an essential role in the antiproliferative activity of bimodular biHD1. The sensitive cell line RL-67 has an ideal response to GQ conformational disordering: biHD1-5’-Δ2G does not show antiproliferative activity at all.

### BiHD1+Ba

Finally, what happened when both GQs of biHD1 would be fixed by a strong chelating cation, similar to barium in the ‘biHD1+Ba’ molecule? It is well known that Ba^2+^ is strongly chelated into the central cavity of GQ and stiffens it [29]. Moreover, the conformation of ‘HD1+Ba’ is not able to interact with the target protein (thrombin), normally interacting with HD1 [25]. 

The CD spectrum of biHD1+Ba differs significantly from the spectra of other derivatives of biHD1 (Figure 1). Similar to HD1 [25] the magnitude of biHD1+Ba CD spectrum maximum of antiparallel GQ is significantly increased (by 1.5 times) and shifted to 300 nm.

For almost all cell lines, there is no antiproliferative effect at all concentrations of biHD1+Ba (Appendix A). This observation supports an idea that a certain GQ conformation or conformational flexibility of GQ is required for the antiproliferative activity of biHD1, which is in line with the behavior of other GQs+Ba and their targets.

## 4. Materials and Methods

### 4.1. Oligodeoxyribonucleotides (ODN)

DNA oligonucleotides were synthesized and HPLC-purified by Evrogen Ltd. (Moscow, Russia).

The nucleotide sequences of ODNs are provided in Table 1. Salts for buffers were of analytical grade purity from MP Biomedicals (France). The 100 μM ODN stock solutions were prepared in 10 mM NaH_2_PO_4_/Na_2_HPO_4_, 10 mM KCl and 140 mM NaCl with pH of 7.5 (PBS). The ODN solutions were subjected to annealing procedure: Heating to 95 °C, then slow cooling to 25 °C. A two-fold molar excess of barium chloride was added to biHD1 to prepare ‘biHD1+Ba’ samples. 

### 4.2. Circular Dichroism Spectroscopy

Circular dichroism spectroscopy (CD) samples of oligonucleotides were prepared with 2.5 μM ODN (5 μM for HD1) in phosphate-buffered saline (10 mM NaH_2_PO_4_/Na_2_HPO_4_, 10 mM KCl, 140 mM NaCl, pH 7.5, PBS). CD spectra were registered on CD spectrometer CHIRASCAN (Applied Photophysics, Leatherhead, Surrey, UK) and dichrograph MARK-5 (Jobin-Yvon, Bensheim, Germany). Quartz cuvettes with an optical path length of 1 cm were used. CD spectra were registered in the wavelength range from 220 to 340 nm at the scan rate of 120 nm min^−1^, with a response of 2 s at 2.0 nm bandwidth, and normalized by subtraction of the background scan with buffer only. The temperature was kept constant at 20 °C with a thermoelectrically controlled cell holder. The CD signal was recalculated as molar circular dichroism, Δɛ (cm^−1^ M^−1^).

All data processing and calculations were performed using the OriginPro 8.0 software (Origin Lab Corporation, Northampton, MA, USA).

### 4.3. Cell Lines and MTT Assay

Cell lines are listed in Table 1. The RL-67 and mS cell lines were kindly provided by N.N. Blokhin National Medical Research Center of Oncology, Russia. The other cell lines were purchased from the Institute of Cytology, Russian Academy of Science, St.Petersburg, Russia. Cells were cultured in Dulbecco’s modified medium (DMEM, Gibco, USA) supplemented with 10% fetal bovine serum (FBS, Thermo Fisher Scientific, Waltham, MA, USA) at 5% CO_2_ at 37 °C.

The cell viability was measured by MTT test using colorimetric measurements of metabolic activity of viable cells. The cells were seeded into 96-well plates with a density of 3 × 10^3^ cells per well and incubated for 24 h at 37 °C. Then, the medium was replaced with a 50% solution of medium with ODN at various concentrations (0.10, 1.0 and 10 μM) or phosphate buffer only. The cells were incubated for three days. Then, 20 μL of 5 mg/mL MTT reagent (3-(4,5-dimethylthiazol-2-yl)-2,5-diphenyltetrazolium bromide) (Sigma-Aldrich, USA) was added to each well, and the incubation was continued for 2 h at 37 °C. The medium was removed from the wells and 60 μL/well of DMSO (PanEco Ltd., Russia) was added, stirred for 15 min, and the absorbance at 540 nm was measured using Tecan plate reader (Tecan Group Ltd, Männedorf, Switzerland). The data were normalized to MTT results for the cells without ODN; the results represent the mean ± SEM of *n* = 5 repeats.

### 4.4. Immunohistochemistry

U87 and MCF7 cells were grown in a 4-well plate, 1 × 10^5^ cells per well. Cells were incubated with biHD1 solutions with final concentrations of 0.10, 1.0 and 10 μM in PBS or PBS without ODN in 50% cell medium for 72 h at 37 °C and 5% CO_2_. Cells were washed three times with PBS, and the solution of 3.7% formaldehyde in phosphate buffer was added to each well for fixation, the cells were washed three times with PBS. The primary antibody to the Ki-67 protein proliferation marker (NCL-Ki67p, Novocastra Laboratories, Newcastle upon Tyne, UK) with 0.1% Triton×100 was added to the cells for 1 h at 20 °C. Cells were washed with PBS three times, then secondary antibody conjugated with the fluorescent dye Cy2 (Cy2-conjugated AffiniPure donkey anti-rabbit, cat.no 711-225-152, Jackson ImmunoResearch, Cambridgeshire, UK) was added to the cells for 1 h. After washing three times with PBS, the cells were tinted with a bisbenzimide nuclear fluorescent dye (H33342, Sigma-Aldrich Corp., St. Louis, MO, USA). The cells were photographed using an Olympus IX81 inverted fluorescence microscope (Olympus Corporation, Tokio, Japan) equipped with a D-70 digital camera (Leica Camera AG, Wetzlar, Germany) applying two modes to register the fluorescence of the Cy2 dye and bisbenzimide. For each well, six representative images were randomly captured. The percentages of Ki-67 positive cells were quantified by a masked researcher. For every captured area, the amount of Ki-67 positive cells (stained with Cy2) and the total amount of cells (stained with bisbenzimide) were counted. Ki-67 negative cells were not stained with Cy2. In total, for each dose of biHD1 more than 600 cells were analyzed.

### 4.5. Statistical Analysis

Statistica 13.0 software (StatSoft, Tulsa, OK, USA) was used for data analysis. The data are shown as ± SD. Analysis of variance was used to evaluate statistical differences between groups. The χ2 test was applied to evaluate the difference between the enumeration data of different groups. The Mann–Whitney test was used to determine the mean number of Ki-67-positive cells per field. A statistically significant difference was assumed for *p* < 0.05.

## 5. Conclusions

Building a novel molecular construct by combining several functional units (modules) is a perspective direction of a molecular design, and oligonucleotides are very suitable molecules to perform these tasks. Development of nucleic acids therapeutics requires intensification of this direction as well. 

Here, a feasibility of building of antiproliferating bimodular construct by covalently linking two antiproliferative DNA antiparallel G-quadruplex (GQ) 15-mers (HD1) have been demonstrated. Antiproliferative activities of bimodular GQ biHD1 and its derivatives were tested using eight different human cancer cell lines: Lung cancer RL-67, central nervous system cancer U87, epithelioid cervix carcinoma HeLa, colon cancer HCT116, breast adenocarcinoma MCF7, melanoma mS, prostate cancer PC3, and a control non-malignant human embryonic fibroblast—hEF. Antiproliferative effects of biHD1 vary for different cell lines; therefore, there is a lack of unique mechanism. BiHD1 does not inhibit proliferation of control non-cancerous hEF. BiHD1 exhibits a dose-dependent antiproliferative activity for RL-67 and U87. The study of derivatives of biHD1 for RL-67 and U87 revealed a strict structure-activity correlation of GQ folding and antiproliferative activity.

These initial results are bases from where to further explore the antiproliferative properties of GQs and to study the following topics: How does GQ penetrate into the cells; molecular and cellular mechanisms; what is a target; and others. This research provides an experimental support for the development of new GQ-based antiproliferative agents either as GQ itself or as a carrier.

## Figures and Tables

**Figure 1 molecules-24-03625-f001:**
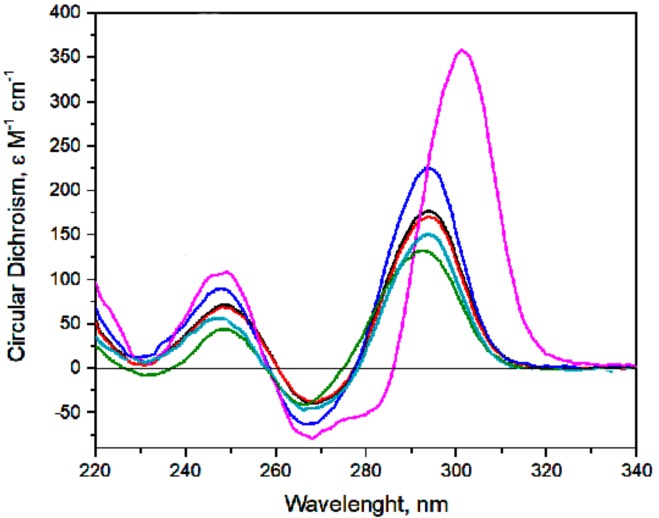
CD spectra of 2.5 µM biHD1 and ODNs under the study in 10 mM NaH_2_PO_4_/Na_2_HPO_4_, 10 mM KCl, 140 mM NaCl, pH 7.5 at 20 °C; biHD1—blue line, biHD1-C3—black line, biHD1-T4A,T20A—red line, biHD1-5’-Δ2G—green line, biHD1+Ba—purple line, HD1—light blue line.

**Figure 2 molecules-24-03625-f002:**
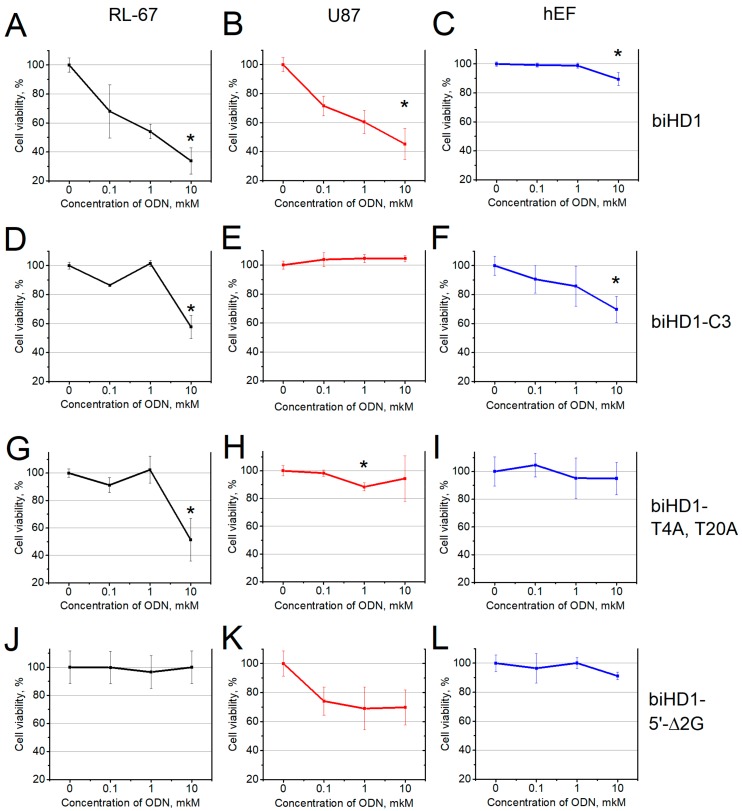
Cell viability according MTT test for oligonucleotides: (**A**–**C**)—biHD1; (**D**–**F**)—biHD1-C3; (**G**–**I**)—biHD1-T4A,T20A; (**J**–**L**)—biHD1-5′-Δ2G; results are presented for human cancer cell lines RL-67 (**A**,**D**,**G**,**J**), U87 (**B**,**E**,**H**,**K**), and embryonic fibroblasts (**C**,**F**,**I**,**L**) after 72 h incubation with different concentrations of oligonucleotides: Control experiment without oligonucleotides, 0.10, 1.0, and 10 μM. Data are presented as mean ± S.D. * marks *p* < 0.05 compared with control (w/o oligo).

**Figure 3 molecules-24-03625-f003:**
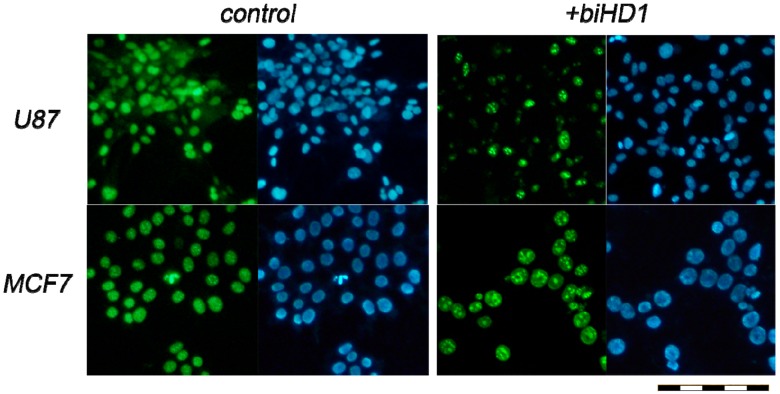
Fluorescence images of U87 cells (up) and MCF7 cells (bottom) incubated with 10 μM biHD1 (right panel), and with no oligonucleotide added (control left panel) for 72 h. Proliferating cells were stained with fluorescent Ki-67-antibody, green images; all cells were stained with bisbenzimide nuclear fluorescent dye, blue images. Scale bar for all images = 100 μm.

**Table 1 molecules-24-03625-t001:** Sequences of DNA oligonucleotide biHD1 and its derivatives. The changes are shown in bold; **pr**—1,3-propanediol residue.

Number	Name	ODN Nucleotide Sequence
1	HD1	GGTTGGTGTGGTTGG
2	biHD1	GGTTGGTGTGGTTGG**T**GGTTGGTGTGGTTGG
3	biHD1-C3	GGTTGGTGTGGTTGG**pr**GGTTGGTGTGGTTGG
4	biHD1-T4A,T20A	GGT***A***GGTGTGGTTGG**T**GGT***A***GGTGTGGTTGG
5	biHD1-5′-∆2G	TTGGTGTGGTTGG**T**GGTTGGTGTGGTTGG
6	biHD1+Ba	GGTTGGTGTGGTTGG**T**GGTTGGTGTGGTTGG

**Table 2 molecules-24-03625-t002:** Cell viability after treatment of human cancer cell lines and control cell line with ‘standard’ concentration of oligonucleotides, 10 μM. The cell line is listed according to the cell viability after biHD1 treatment. The viability values with less than 70% are in bold. All data could be found in the Appendix A. In the biHD1-C3 column, * marks the data for the single-modular HD1.

Cell Line	Cell Viability, %				
	biHD1	biHD1-C3	biHD1-T4A, T20	biHD1-5′-∆2G	biHD1+Ba
RL-67	**34**	**58**	**51**	100	103
U87	**45**	104/103 *	94	**66**	96
HeLa	**57**	**70**	91	74	86
HCT116	**58**	96	105	74	100
MCF7	**64**	90	102	85	87
mS	77	96	93	93	98
PC3	78	131	91	85	76
hEF	90	70/87 *	95	91	101

**Table 3 molecules-24-03625-t003:** List of the human cancer cell lines and the control cell line.

Number	Name	Cell Line	Reference
1	RL-67	Lung cancer	[36]
2	U87	Central nervous system cancer	[37]
3	HeLa	Epithelioid cervix carcinoma	[38]
4	HCT116	Colon cancer	[39]
5	MCF7	Breast adenocarcinoma	[40]
6	mS	Melanoma	[41]
7	PC3	Prostate cancer	[42,43]
8	hEF	Human embryonic fibroblasts	[44]

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
