# Peer review of "Bimodular Antiparallel G-Quadruplex Nanoconstruct with Antiproliferative Activity"

_molecules, 2019, doi:10.3390/molecules24193625_

Round 1
Reviewer 1 Report
The submitted manuscript (molecules-572626) entitled: “Minimal bimodular G-quadruplex nanoconstruct with antiproliferative activity” has been already reviewed by me twice. First of all I do not understand the meaning of the word “minimal” in the title. Is it necessary?
The manuscript has been partially revised after the second revision - Fig. 4 and Fig 5 were added and the antiproliferative results for monomodal HD1 were provided for two cell lines (unfortunately not for all cells studied – referring to results of other Authors may be risky because of experimental condition differences). My suggestion to measure melting profiles (Tm determination) for all investigated GQs was also only partially fulfilled by Authors and they argued that the screening study against cell lines is their priority goal. Nevertheless, they speculated on the stability of GQs and it potential effect on anticancer activity. In my opinion, known GQ folded fraction for all oligonucleotides tested (especially for the truncated one would be helpful in discussion of results. Presented results have some practical value (potential anticancer therapy) and rather limited value for the rational drug design.
Reviewer 2 Report
In this project, author examined the antiproliferative activity of biTBA. It will be an interesting article in the field of G-quadruplex only if the following questions have been addressed.
Major points:
1.DNA is negative charge therefore they could not enter the cells through diffusion. Could they internalize into the cells? Fluorescence labeled oligo may help to examine the target location.
2. Section 3.1 only describe the proposal DNA for testing. Authors did not mention the reasons of choosing those DNA sequence. What is the rational for making biHD1 from HD1? Why biHD1+Ba? What is the hypothesis of using Ba2+ stabilized biHD regarding to antiproliferative activity?
3. What is the the purpose and conclusion of experiment in figure 2? what is the meaning of "more cooperative behavior of GQ melting"?
4. Author should use one-way analysis of variance (ANOVA) to determine whether there are any statistically significant differences in Figure 3.
5. Author should investigate the IC50 at least for bi HD1.
6. How to define Ki67 positive cells? How about measure the average whole cell intensity of green? Author should also use a positive control for this experiment.
7. The method section is not complete. For example,
Authors only mentioned the source of RL-67 and mS cell lines. Authors should mention the source of other cell line.
Authors should mention the product code of primary Ki67 antibody and the source of secondary antibody.
8. does the antiproliferative activity depends on the formation of G-quadruplex? what happen if a mutation sequence which cannot form G4, was used.
Minor points:
1. "HD1" is is widely known as thrombin binding aptamer (TBA). Why authors name is as HD1 in this manuscript.
2. The word "0, 0.1, 1, 10 below the x axis is too crowd"
Reviewer 3 Report
The main focus of the manuscript submitted by Antipova et al. is to demonstrate the feasibility of building multimodular structures by combining two G-quadruplex modules and create functional oligonucleotides. Their antiproliferative activities were screened on a selection of human cancer cell lines of different nature. Very promising results were obtained which makes this study important contribution to the field. However, structural part of the paper that is based mostly on CD spectra is over-interpreted and some (minor) corrections are needed.
Hypothesis on why 'HD1 has antiproliferative activity, then it is of value to use this GQ as a building block for assembly of multimodular constructs' needs to be elaborated more extensively and presented more clearly.
p.4; what experimental evidence is used to state that barium cation binds G-quadruplexes much stronger than potassium cation?
It is not a good practice to refer to one of the two loops of structure of HD1 as the 'bottom' loops. This should be corrected in several places in the manuscript.
CD spectrum in the presence of Ba2+ ions shows different shape and maxima at different wavelengths which may indicate different structures. This should be clearly stated.
p. 6; Sentence 'Destabilizing two GQs of biHD1 by replacing a single thymidine linker with a more flexible three-methylene bridge provides GQs with more degrees of freedom, so the whole structure becomes disordered, as can be seen from a decrease in the amplitude of the maximum antiparallel GQ in the CD spectrum of biHD1-C3 (Fig. 1).' needs to be rephrased. Peaks in CD spectra show only slightly lower intensity and in no way suggest 'disorder' of structure.
p. 1, line 38: References to position of potassium cation between two stacked G-quartets references no. [10, 11] should be corrected to cite a work published in Org. Biomol. Chem. 2009, 7, 4677, which was the first to demonstrate position of a cation between G-quartets in HD1.
p. 9, line 249; there must be a typo in 'about 10% was observed with 20 μM HD1' as no results at 20 micro-M concentration are presented in Figure 5, assuming that data in the figure are correct.
p. 9; There is insufficient evidence to state that 'initial structure of GQ is affected, either indirectly or directly'.
What data in this manuscript support a statement that two G-quadruplexes were pulled apart in biHD1 analogues? Are two G-quadruplexes really formed?
Structures on the right side of Figure 3 are not supported with sufficient data to allow for almost atomic resolution and may be highly misleading, and should as such be removed.
Round 2
Reviewer 2 Report
Authors replied all my questions and I think this manuscript is interesting for the audience of Molecules.
This manuscript is a resubmission of an earlier submission. The following is a list of the peer review reports and author responses from that submission.
Round 1
Reviewer 1 Report
The main weakness of the paper is that the biological assay used was only proliferation and no other parameters. To publish a study in a biological paper you must use other methods such as apoptosis assays, migration and so on.
Other points:
The Results section should be divided to sub chapters.
I would summarize all the results in one table.
The conclusions should include the definition of the best compound synthesizes by the authors, taking into account the effect on the viability of the control cells.
At the beginning of the introduction it is not clear what receptor the authors meant.
Legend to Fig. 3: replace amount with concentration.
There are many English grammar mistakes.
Author Response
We are grateful to the reviewers for their attentive attitude to our manuscript, and valuable comments that helped us significantly improve the manuscript.
Responce to the reviewer 1
We have clarified research design, in particular by expanded the descriptions of derivatives of biHD1.
3.1. Oligonucleotide design.
The nucleotide sequences of biHD1 and its derivatives are shown in Table 1.
biHD1 is a 31-mer which was created by covalently joining two 15-mer HD1s via a single thymidine residue T16 (previously known as RA-36 for thrombin) [23].
biHD1-C3 has a more flexible three-methylene linker instead of T16 between two HD1s, which is supposed to pull apart GQs, reducing mutual influence.
biHD1+Ba is the original biHD1 with Ba2+ cations in the central cavities of both GQs, which was made by using Ba2+ solution instead of conventional K+. Barium cation binds GQ much stronger than potassium cation and highly stabilizes, stiffens GQ. In the case of monomodular HD1, excessive stabilization reduces the mobility of bottom TT loops and, as a consequence, obliterates binding with thrombin [33].
In biHD1-T4A,T20A, the second thymidines in the first TT loops of both GQs of biHD1 were replaced with adenines: T4A and T20A, respectively. In the original structure, a pair of thymine bases T4-T13 are in stacking interactions with the lower G-quartet of the chair-like structure. Since the adenine base is larger and interacts with the thymine base differently, T4A,T20A replacements should destabilize both putative GQs. For example, T4A replacement in the original monomodular HD1 resulted in a loss of aptamer binding to thrombin [34].
biHD1-5'-Δ2G is a 5'-truncated version of biHD1. This deletion of the first two Gs of the first GQ should not allow a proper folding of the first GQ [35].
The main weakness of the paper is that the biological assay used was only proliferation and no other parameters. To publish a study in a biological paper you must use other methods such as apoptosis assays, migration and so on.
We completely agree with the reviewer that the variety of methods could be applied to study the functional effects of oligonucleotides, not just MTT-test.
The goal of research is screening of variants of biHD1 for a number of cell lines. And for this purpose MTT-test is of primary choice. After having a suitable cell line it could be possible to study in more details by the methods suggested by the reviewer.
The Results section should be divided to sub chapters.
Results section has been divided to sub chapters.
I would summarize all the results in one table.
We agree, the results have been summarized according to the comment.
The conclusions should include the definition of the best compound synthesizes by the authors, taking into account the effect on the viability of the control cells.
The conclusions has been changed and corrected according to the reviewer's comments.
5. Conclusions
Here, we demonstrate the feasibility of building multimodular structures by combining GQ modules and creating functional oligonucleotides with antiproliferative activity. Antiproliferative activities of bimodular G-quadruplex biHD1 and its derivatives were screened on a selection of human cancer cell lines of different nature, in order to identify sensitive cell lines. Like the original monomodular HD1, the bimodular biHD1 exhibits antiproliferative activity. Antiproliferative effect varies for different cell lines, therefore this step of screening is quite necessary for any study of antiproliferative activity of a particular GQ, and simple use of only a specific cell line of interest is uninformative.
Antiproliferative activity of biHD1 for some cell lines depends on GQ folding. For example, disruption of the first GQ by removal of the first two Gs, as in biHD1-5’-Δ2G, when treating the reference cell line RL-67, completely abolished the antiproliferative activity. This is consistent with the existing paradigm that the putative mechanism of antiproliferative activity of GQ may depend both on the specific structure of GQ and on the specific nature of cell line. This emphasizes the importance of screening experiments described in this publication.
At the beginning of the introduction it is not clear what receptor the authors meant.
The sentence has been changed:
The mechanism of this effect is not yet known, though it may be the result of a lot of different events: oligonucleotide binding to the cell membrane, penetration into cell, specific binding and inhibition of GQ-binding proteins that could be pleiotropic [5] and playing an essential role in cell proliferation, such as participating in oncogene transcription or signal transduction [5-9].
Legend to Fig. 3: replace amount with concentration.
The legend has been corrected.
There are many English grammar mistakes.
After major revision according to the reviewer’s comments the manuscript has been checked by English speaking person.

Reviewer 2 Report
The submitted manuscript entitled “Bimodular G-quadruplex nano-construction with antiproliferative activity” reports the antiproliferative activity of the TBA-related oligonucleotides using a series of cancer cell lines and a normal cell line. The studied oligonucleotide family was based on the bi-modal oligonucleotide (bi-HD1) possessing two G-quadruplex units with the sequence of thrombin binding aptamer (denoted as TBA or HD1). Authors tested five bi-HD1 derivatives and tried to find a correlation between their structural characteristics and antiproliferative activity. Oligonucleotides exhibited diverse antiproliferative activity towards particular cell lines without any obvious correlation. According to Authors, the antiprofiferative activity of tested derivatives is related to their G4 folding abilities. Since The antiproliferative effect varied for different cell lines, Authors speculated that it was based on different mechanisms depending on the specific cell line. The results are interesting and were clearly presented but some issues listed below should be addressed before publication of manuscript.
1) It is not clear whether bi-modal GQ is more effective than mono HD1. Authors should include results for antiproliferative effect of monomer HD1 towards tested cell lines. This mono-GQ oligo will serve as a reference to demonstrate that studied bimodal systems are better antiproliferative agents. On the other hand such results may help in the discussion related to structure-activity issue.
2) The five bi-HD1 derivatives should be more extensively characterized. I would suggest to carry out thermal melting experiments to prove assumptions (speculations) on GQ stability that Authors presented in the discussion of results (e.g., on page 8: A double replacement of two thymines for adenine destabilizes both GQ in biHD1-4,20T/A oligonucleotide.”). Low intensity of CD signal is not sufficient to conclude on low stability.
3) Citation of references should be revised. For example, ref. [23] is between [32] and [34], in page 3, sentence: “… biHD1 molecule was demonstrated previously, for example, during thermal melting [34].” – no melting data in ref [34].
4) Captions for figure 1 are not sufficient – experimental details are missing.
5) A sentence on page 6: “…In particular, for some cell lines, namely: U87, HCT116, MCF7, mS, the responce is absent at all. (Fig 2C, Fig. S1H).” Refers incorrectly to Fig S1H
6) A table with sequences of particular oligonucleotides should be included (e.g., in ESI)
Author Response
Response to the reviewer 2
The manuscript has been major revised. The design of oligonucleotides for the study has been carefully described: a separate chapter describes design of biHD1 derivatives.
The conclusions have been also changed.
Authors tested five bi-HD1 derivatives and tried to find a correlation between their structural characteristics and antiproliferative activity. Oligonucleotides exhibited diverse antiproliferative activity towards particular cell lines without any obvious correlation.
The main focus of the research is screening of antiproliferative activity of biHD1 and its derivatives on various cell lines to identify the most sensitive cell line. The most obvious correlation of GQ structure and the activity is for RL-67 cell line.
The graphical presentation of the data has also been changed. In addition, the summary table of the results has been inserted into the text.
According to Authors, the antiprofiferative activity of tested derivatives is related to their G4 folding abilities. Since The antiproliferative effect varied for different cell lines, Authors speculated that it was based on different mechanisms depending on the specific cell line. The results are interesting and were clearly presented but some issues listed below should be addressed before publication of manuscript.
1) It is not clear whether bi-modal GQ is more effective than mono HD1. Authors should include results for antiproliferative effect of monomer HD1 towards tested cell lines. This mono-GQ oligo will serve as a reference to demonstrate that studied bimodal systems are better antiproliferative agents. On the other hand such results may help in the discussion related to structure-activity issue.
The monomodular HD1 had been extensively studied, and the antiproliferative effect of HD1 has been discussed in the manuscript (the literature is listed below).
In addition, as it has been discussed in the manuscript, GQs with different structures have different effects for different cell lines. The mechanism is not known in detail to every particular GQ, therefore it makes sense to screen sensitive cell line to biHD1 first, and then compare it with HD1 for this cell line applying different tests, not just MTT.
13. Dapić, V.; Abdomerović, V.; Marrington, R.; Peberdy, J.; Rodger, A.; Trent, J.O.; Bates, P.J. Biophysical and biological properties of quadruplex oligodeoxyribonucleotides. Nucleic Acids Res. 2003 31(8), 2097-107.
16. Scuotto, M.; Rivieccio, E.; Varone, A.; Corda, D.; Bucci, M.; Vellecco, V.; Cirino, G.; Virgilio, A.; et al. Site specific replacements of a single loop nucleoside with a dibenzyl linker may switch the activity of TBA from anticoagulant to antiproliferative. Nucleic Acids Res. 2015 43(16), 7702-16. doi: 10.1093/nar/gkv789.
17. Esposito, V.; Russo, A.; Amato, T.; Varra, M.; Vellecco, V.; Bucci, M.; et al. Backbone modified TBA analogues endowed with antiproliferative activity. Biochim Biophys Acta - Gen Subj. 2018 1861(5 Pt B), 1213-1221.
18. Esposito, V.; Russo, A.; Amato, T.; Vellecco, V.; Bucci, M.; Mayol, L.; et al. The “Janus face” of the thrombin binding aptamer, Investigating the anticoagulant and antiproliferative properties through straightforward chemical modifications. Bioorg. Chem. 2018 76, 202–209.
2) The five bi-HD1 derivatives should be more extensively characterized. I would suggest to carry out thermal melting experiments to prove assumptions (speculations) on GQ stability that Authors presented in the discussion of results (e.g., on page 8: A double replacement of two thymines for adenine destabilizes both GQ in biHD1-4,20T/A oligonucleotide.”). Low intensity of CD signal is not sufficient to conclude on low stability.
Thermal melting experiments have been done, and CD-melting curves have been introduced on Fig.2.
3) Citation of references should be revised. For example, ref. [23] is between [32] and [34], in page 3, sentence: “… biHD1 molecule was demonstrated previously, for example, during thermal melting [34].” – no melting data in ref [34].
Citation of references has been revised and corrected.
4) Captions for figure 1 are not sufficient – experimental details are missing.
The captions of figures 1 and 2 have been extended with experimental details, also the experimental details have been added into Materials and Methods section.
5) A sentence on page 6: “…In particular, for some cell lines, namely: U87, HCT116, MCF7, mS, the responce is absent at all. (Fig 2C, Fig. S1H).” Refers incorrectly to Fig S1H
The references to figures have been corrected. The graphical presentation of the data has also been changed.
6) A table with sequences of particular oligonucleotides should be included (e.g., in ESI)
The table with the nucleotide sequences of the oligonucleotides has been inserted into the text.

Round 2
Reviewer 1 Report
The authors revised the manuscript according to my suggestions; however they are not willing to test their compounds by other methods as well (although they agree with me……). Therefore I suggest adding a paragraph in the discussion saying that the lack of other methods is a limitation in the study and that other methods will be employed in further studies.
Reviewer 2 Report
The submitted revised version of manuscript (molecules-488269) under a changed title: “Minimal bimodular G-quadruplex nanoconstruct with antiproliferative activity” has only partially addressed issues pointed out in my review (why minimal?).
My comments #1 and 2 are crucial for discussion of activity mechanism of these GQ-related oligonucleotides. The reply of Authors that many other research groups have studied HD1-related systems is not convincing. Authors tested many cancer lines and for some of them related data for HD1 are missing in literature. Moreover, one should provide evidence that modification of an original molecule (HD1) gives more efficient or advantageous drugs. Otherwise, reported results may be regarded as negative ones. On the other hand, data for HD1 should be useful in discussion of mechanism. Authors speculated on the effect of GQ numbers in molecule on the activity (e.g., biHD1 vs. biHD1-5′-Δ2G) but comparison with mono GQ (HD1) seems to be more straightforward.
Another aim of the reported study seemed to be investigation of the relationships between the structural features of quadruplex-forming oligonucleotides, their biophysical properties (CD spectra) and their antiproliferative activity. The major conclusion was that such simple relationships did not exist. It is not surprising conclusion since GQ-protein binding (and thus biological activity) is mediated by recognition of the specific shape of the quadruplex construct, which will depend on many factors including thermal stability of GQ structure. For example, the melting profiles in Fig. 2 suggest Tm values (not reported) below 40oC, which means that at antiproliferative activity experiments (37 oC) only ca. 50% of oligonucleotide was in a folded form. This factor (and Tm values for other drugs) was completely ignored by Authors.
Concluding, manuscript should not be published without the suggested additional results (antiproliferative activity data for HD1 and melting data (including Tm values) for all five oligonucleotides tested).